# Narrative Review: Predictive Biomarkers of Tumor Response to Neoadjuvant Radiotherapy or Total Neoadjuvant Therapy of Locally Advanced Rectal Cancer Patients

**DOI:** 10.3390/cancers17132229

**Published:** 2025-07-03

**Authors:** Joao Victor Machado Carvalho, Jeremy Meyer, Frederic Ris, André Durham, Aurélie Bornand, Alexis Ricoeur, Claudia Corrò, Thibaud Koessler

**Affiliations:** 1Translational Research Center in Onco-Hematology, Department of Medicine, Faculty of Medicine, University of Geneva, 1205 Geneva, Switzerland; joao.carvalho@unige.ch (J.V.M.C.); claudia.corro@unige.ch (C.C.); 2Swiss Cancer Center Léman, 1005 Lausanne, Switzerland; 3Department of Oncology, Geneva University Hospital, 1205 Geneva, Switzerland; 4Division of Digestive Surgery, University Hospitals of Geneva, Rue Gabrielle-Perret-Gentil 4, 1205 Geneva, Switzerland; 5Department of Radiation Oncology, University Hospitals of Geneva, 1205 Geneva, Switzerland; andre.durham@hug.ch; 6Pathology Department, Geneva University Hospitals, 1205 Geneva, Switzerland; 7Service of Radiology, Geneva University Hospital, 1211 Geneva, Switzerland; alexis.ricoeur@hug.ch

**Keywords:** rectal cancer, total neoadjuvant treatment, short-course radiotherapy, radiotherapy, tumor response, pCR, TRG, biomarker

## Abstract

Locally advanced rectal cancer treatment consists of neoadjuvant treatment (NAT) followed by surgery. Neoadjuvant treatments can include chemoradiotherapy (CRT), short-course radiotherapy (SCRT), radiotherapy (RT) or total neoadjuvant treatment (TNT). Each of these treatment modalities impact a patient’s quality of life. Finding predictive markers of tumor response enabling a personalized treatment approach is of high interest. Most of the research on biomarkers has focused on tumor response following CRT. This review aims at gathering current knowledge of biomarkers predicting response to neoadjuvant treatments other than CRT, such as SCRT, RT and TNT regiments.

## 1. Introduction

Rectal cancer (RC) represents one third of all colorectal cancers with 125’000 incidental cases in 2017 and a mortality of 4–10 deaths/100’000 individuals [1]. In Europe, locally advanced RC (LARC), defined as stage II and III RC, is the most prevalent stage at diagnosis, representing 42–44% of patients at diagnosis according to the EUROCARE-5 study [1,2]. According to European and American medical oncology guidelines, LARC should receive neoadjuvant treatment (NAT) with either chemoradiotherapy (CRT) or short-course radiotherapy (SCRT) followed by curative surgery for the least advanced cases [1,3]. More advanced disease, typically mrT3 threatening or invading the mesorectal fascia, mrT4 and/or mrN2, are offered intensified treatments with additional induction and/or consolidations chemotherapies—called total neoadjuvant treatment (TNT) [4,5,6,7].

In LARC, current neoadjuvant treatments lead to pathological complete response (pCR) in 6 to 39% of patients [8,9]. Being able to predict tumor response to NAT is of utmost importance, as it could determine which patients would benefit from it and spare the side effects for those who would not [9,10]. Furthermore, clinical complete response (cCR) or near cCR used as a surrogate of pCR can lead to a *watch-and-wait* strategy, allowing organ preservation and avoiding surgical complications [7,11].

To this day, no biomarkers are available in clinics to predict the tumor response to NAT in LARC patients [1,12,13,14,15]. Furthermore, research has focused on the tumor response to CRT; while the treatment landscape has evolved in the past years, only a scarce number of studies have searched for predictive biomarkers in patients treated with SCRT or TNT. The aim of our review is to clarify whether there are predictive biomarkers of tumor response in LARC patients treated with SCRT, TNT or other radiotherapy (RT) alternatives to CRT.

## 2. Materials and Methods

We performed a literature search in EMBASE and MEDLINE on the 28 March 2025 using the following keywords: “locally advanced rectal cancer”, “rectal cancer”, “LARC”, “short-course radiotherapy”, “SCRT”, “total neoadjuvant treatment”, “TNT”, “neoadjuvant radiotherapy”, “biomarker”, “liquid biopsy”, “gene expression”, “protein marker”, “tumor response”, “tumor regression”, “tumor regression grade”, “pathological complete response”, “tumor remission”, “predictive value” and “prediction”, with the Boolean AND/OR.

Included full-text articles involved LARC patients (stage II and III RC) that were treated with neoadjuvant SCRT (5 × 5 Gy), TNT or other neoadjuvant RT schemes. Studies on patients undergoing standard CRT or chemotherapy alone were excluded. In cases where a study had multiple subgroups based on treatment, only the subgroups meeting the inclusion criteria were considered.

All types of biomarkers (circulating, proteomic, genomic or radiomic) were considered.

All the articles needed to have the tumor response as the main or secondary outcome. PCR, tumor regression grade (TRG), the Response Evaluation Criteria in Solid Tumors (RECIST) or tumor downstaging were accepted assessment methods of tumor response. Studies evaluating only overall survival (OS) and/or disease-free survival (DFS) were not included.

## 3. Results

### 3.1. Literature Searches

The literature search identified two hundred and fourteen articles. Six additional records meeting the inclusion criteria were identified through reviews and citations. After removing duplicates (*n* = 5), two hundred and fifteen articles were screened. One hundred and seventy-three articles were excluded based on title and abstract. Forty-two full-text articles were assessed for eligibility and twenty-one articles were excluded based on the inclusion and exclusion criteria. In total, twenty-one studies were included (Figure 1).

Seven studies investigated biomarkers in association with SCRT [16,17,18,19,20,21,22]. Eight studies investigated biomarkers in association with other RT regimens [23,24,25,26,27,28,29,30]. Six studies investigated biomarkers in association with TNT [31,32,33,34,35,36]. An overview of the selected articles is available in Table 1.

### 3.2. Circulating Markers

Circulating markers were investigated in seven articles [17,20,22,28,31,34].

#### 3.2.1. CEA

Carcinoembryonic antigen (CEA) is a glycoprotein expressed on epithelial cell membrane [37]. It loses polarization in tumor cells and may be measured in the blood circulation. It is measured in RC to assess prognosis and detect tumor relapse or progression after surgery [1].

Two studies considered CEA as a predictive biomarker of tumor response in LARC patients [28,31]. Chapman et al. found that low pre-treatment levels of CEA (<3 ng/L) were associated with pCR in LARC patients receiving TNT (*p* = 0.003) [31]. In patients receiving RT, Wang et al. found that lower pre-treatment levels of CEA (<5 ng/L and <10 ng/L) were associated with downstaging (*p* = 0.001) and particularly with T downstaging (*p* = 0.00014) [28].

#### 3.2.2. Liquid Biopsy

Liquid biopsies consist of the analysis of non-solid tissues for tumor early diagnosis, detection of tumor relapse after surgery, or identification of treatment targets with the measure of circulating tumor cells (CTC) or nucleic acids, usually circulating free DNA (cfDNA), circulating tumor DNA (ctDNA) or microRNA (miRNA) [38,39].

cfDNA can be measured in circulation in oncologic, autoimmune or inflammatory conditions following tissue damage or after being actively secreted [40,41]. One study considered the cfDNA predictive value [20]. Truelsen et al. showed that, for patients treated with SCRT, a reduction in the median cfDNA value below the 75th percentile following SCRT was associated with pCR (*p* = 0.001) [20].

ctDNA is a fragmented part of tumor DNA found in the blood circulation [42]. Two studies investigated ctDNA levels in LARC patients undergoing TNT but could not show any association between ctDNA levels and pCR [34,36].

#### 3.2.3. Flow Cytometry

Flow Cytometry (FC) characterizes cells at a single-cell level from blood, bone marrow or tissue [43]. Immunophenotyping is one of the main applications of FC but it can also be used for cell cycle, protein or antigen response analysis.

Two articles studied FC for patients treated with SCRT [17,22].

Gasinska et al. performed FC on LARC biopsies and surgical specimens of 122 LARC patients [17]. Only the relative MIB-1 LI (ratio between post- and pre-treatment MIB-1 LI) was associated with the tumor response (*p* = 0.005). Furthermore, when separating the results by gender, it only remained statistically significant in the female subgroup (*p* = 0.041).

Napolitano et al. performed FC on fresh blood samples pre-treatment (T0) at week 2 (T2) and week 5 (T5) post-treatment, pre-surgery (T8) and 6–12 months post-surgery (T9/T10) to characterize peripheral Granulocytic myeloid-derived suppressor cells (G-MDSC)(LIN−/HLA-DR−/CD11b+/CD14−/CD15+/CD33+), monocytic myeloid-derived suppressor cells (M-MDSC) (CD14+/HLA-DR−/lowCD11b+/CD33+) and regulatory T cells (Treg) (CD4+/CD25+/FOXP3+/CTLA4+ and CD4+/CD25+/FOXP3+/PD1+). Good responders had higher M-MDSC levels at T5 (*p* = 0.045) and T8 (*p* = 0.012), lower CTLA4+ Treg at T0 (*p* = 0.045) and T5 (*p* = 0.032) and lower PD1+ Treg level at T8 (*p* = 0.043) and T9 (*p* = 0.027) [22].

### 3.3. Tissue Markers

#### 3.3.1. Immunohistochemistry

Immunohistochemistry (IHC) is a testing method used on pathological samples such as biopsies or surgical specimens [44]. The tissue is usually formalin-fixed, paraffin-embedded (FFPE). It allows us to assess specific protein expression in the selected tissue. Eleven studies used IHC to investigate the following markers in LARC: APAF-1, BCL2, CD34, DCC, EGFR, GLUT-1, HER2, hPEBP4, Ki-67, Ku70, mismatch repair status (MMR), MRE11/RAD50/NBS1, nuclear ß-catenin, p21, p53, PD-L1, tumor-infiltrating lymphocytes (TILS) (CD3^-^CD8+), TS and VEGF [18,19,21,22,23,24,25,27,29,30,32].

*BCL-2* is an anti-apoptotic protein, and its overexpression may induce radioresistance in tumor cells [45]. Two studies considered BCL-2 expression [18,30].

Gasinska et al. showed that expression of cytoplasmic BCL-2 in pre-SCRT biopsies was associated with an improved TRG, but only in patients having short breaks before surgery (being defined as has having surgery less than 15 days after SCRT) [18].

Zlobec et al. did not find an association between cytoplasmic BCL-2 expression and tumor response using pCR in pre-treatment samples of patients treated with RT (26 Gy in 4 fractions) [30].

The Epidermal Growth Factor Receptor (EGFR) harbors a tyrosine kinase function, triggering cell growth and survival, a pro-inflammatory reaction and actin remodeling. Mutated EGFR is associated with tumorigenesis and represents a therapeutic target [46].

Zlobec et al. found that higher cytoplasmic and membranous expression of EGFR (>20%) in pre-treatment samples underdoing RT (26 Gy in 4 fractions) were associated with higher pCR rates in univariate (*p* = 0.003) and multivariate analysis (*p* = 0.01) [30].

Human phosphatidylethanolamine-binding protein 4 (hPEBP4) is an anti-apoptotic molecule overexpressed in several solid tumors and is believed to generate radioresistance [47,48]. Qiu et al. found that lower cytoplasmic expression of hPEBP4 (<70%) in biopsy specimens was associated with an improved TRG in univariate (*p* = 0.001) and multivariate analysis (*p* = 0.001) for patients treated with alternative RT (4 × 5 Gy) [25].

Ki-67 is a protein mainly present in the perinucleolar region in the G1 phase and is a marker of cell proliferation that may be highly expressed in tumor cells [49]. Gasinska et al. found that higher nuclear Ki-67 expression (cut-off not specified) using the MIB-1 labeling index in pre-treatment biopsies was associated with an improved tumor response according to RECIST criteria (*p* = 0.023) after SCRT with a short break (<15 days) before surgery [18].

Ku70 is a protein that plays a role in DNA repair by binding to double-stranded DNA breaks. Its downregulation has been associated with colitis and the development of colorectal cancer [50].

In LARC patients, Gasisnka et al. found that higher pre-treatment nuclear Ku70 expression (cut-off not specified) was associated with higher tumor downstaging in their female subgroup undergoing SCRT with delayed surgery (>15 days) (*p* = 0.035) [18]. No association with tumor response was found in the whole group.

Nuclear β-catenin is part of the Wnt signaling pathway. It is found in the cytoplasm and cell membrane binded to E-cadherin and plays a role in maintaining normal cell structure. In the presence of the Wnt signal, it translocates to the nucleus and initiates a signaling cascade that leads to cell proliferation [51]. It has also been associated with immune invasion in cancer by inhibiting the differentiation of naïve CD8+ T cells into effector cells [52].

Wang et al. found that higher nuclear expression (>50%) of β-catenin in LARC patients undergoing SCRT was associated with radioresistance and a diminished tumor downstaging in univariate (*p* < 0.001) and multivariate analysis (*p* < 0.001) [21]. On the other hand, having a high expression of membrane nuclear β-catenin was associated with better TRG, although this association was not statistically significant.

Vascular Endothelial Growth Factor (VEGF) is essential in angiogenesis to allow cells to meet their oxygen supply [53]. It is overexpressed in numerous solid tumors and plays a role in their growth, and is therefore a therapeutic target [53]. Zlobec et al. found that low expression of cytoplasmic VEGF (<20%) in pre-treatment biopsies underdoing RT (26 Gy in 4 fractions) was associated with an improved pCR rate in univariate (*p* = 0.004) and multivariate analysis (*p* = 0.009) [30].

No significant associations with tumor response treated with RT, SCRT or TNT were found for APAF-1, CD-34, DCC, GLUT-1, HER-2, MMR status, MRN complex, p21, p53, PD-L1, TILS and TS [18,19,23,24,27,29,30,31,32].

#### 3.3.2. Genomic Markers

The tumors cell genome undergoes point-mutations or alterations such as inversions and translocations that are involved in cell growth, proliferation, metastasis or immune system evasion [54]. Genomic markers are identified by sequencing methods that can be targeted using polymerase chain reaction (PCR) or untargeted using sequencing panels that can include whole genome sequencing [55,56]. It allows tumor profiling and can unveil treatment targets guiding clinical decisions [56].

Six studies looked at genomic markers for possible association with tumor response [23,26,27,31,32,33].

p53 is a tumor suppressor protein that is active after DNA damage and able to induce its repair, cell cycle arrest or cell apoptosis [57]. Mutations in p53 can induce the proliferation of cells with unstable genomes as seen in tumors [57].

Kandioler et al. and Rebischung et al. investigated p53 mutation in LARC patients undergoing RT (25 Gy in 10 fractions and 39 Gy in 13 fractions, respectively) [23,26]. Both found that the absence of mutation was associated with a better tumor T downstaging (*p* < 0.001 and *p* < 0.04, respectively).

Chapman et al. used next-generation sequencing (NGS) to record mutation of KRAS, NRAS, BRAF, PIK3CA, APC, FBXW7, SMAD4 and p53 in biopsies from LARC patients undergoing TNT with CAPOX or FOLFOX followed by CRT [31]. They found that the absence of mutation in p53 (*p* = 0.023) and SMAD4 (*p* = 0.040) was associated with pCR. SMAD4 is a tumor suppressor involved in the TGF- β signaling pathway [58].

Iseas et al. used DNA sequencing with a panel of 72 cancer driver genes in biopsies from LARC patients that underwent TNT with CAPOX followed by CRT [32]. TP53, APC, KRAS, ATM and PIK3CA were the most frequently mutated genes, however only *KRAS* mutations were associated with worse TRG in univariate (*p* = 0.013) and multivariate analysis (*p* = 0.042). KRAS is an oncogene involved in cell proliferation when mutated in several solid tumors [59].

Sclafani et al. analyzed the LCS-6 variant using PCR in biopsies from patients involved in the expert-T clinic trial, a randomized phase II trial that evaluated the efficacy of neoadjuvant CAPOX followed by CRT with or without cetuximab in LARC [33,60]. LCS-6 is a microRNA associated with KRAS expression [61]. They found that carriers of the G allele of LCS-6 had a higher pCR rate (28.1% versus 10.6%; *p* = 0.020) and a better 5-year progression free survival (PFS) rate. Furthermore, this increase in pCR rate was independent of the use of cetuximab.

Lit et al. performed NGS on pre-treatment samples from patients that underwent TNT with CAPOX and camrelizumab followed by CRT [36]. Patients without further disease progression also received consolidation regiment with CAPOX. They found that mutation of the low-density lipoprotein receptor-related protein (LRP1B) was associated with cCR (*p* = 0.03) and a tumor shrinkage of more than 50% (*p* = 0.04). LRP1B is a gene that codes for a cell-surface receptor with diverse biological functions such as receptor-mediated endocytosis and cell signaling [62]. Its role in tumors remains unclear, however its inactivation has been associated with several tumor types.

### 3.4. Radiomic Markers

Radiomic markers represent a non-invasive approach to assess patient’s characteristics through imaging data extracted mainly from CT scans, MRI or PET-CT [63]. Radiomics could constitute surrogate biomarkers of interest for tumor characterization, molecular tumor phenotyping, tumor response or prognosis assessment [63,64]. Several articles have already described the predictive value of radiomics in LARC patients undergoing CRT [65]. Two studies using alternatives neoadjuvant therapies were identified [16,35].

Fusco et al. explored parameters derived from MRI to assess tumor response in LARC patients undergoing SCRT [16]. They showed that the pre-treatment mean value and standard deviation (SD) of conventional biexponential fitting pseudo diffusion (CBFM Dp) (*p* = 0.05 and *p* = 0.03, respectively) as well deviation of the mean value of variable projection pseudo diffusion (VARPRO Dp) (*p* = 0.008) were associated with pCR. They also found that a change in pre- and post-treatment standardized index of shape (SIS) was associated with pCR (*p* < 0.001).

Zhang et al. assessed the predictive value of MRI pre-treatment lymph node (LN) characteristics in patients undergoing TNT with CRT followed by mFOLFOX6 or CapeOX [35]. They found that short-axis LN (<8 mm) (*p* = 0.024), smooth LN borders (*p* < 0.001), homogeneous signal intensity (*p* < 0.01) and their radiomic score based on five radiomic features (contrast, long run high gray level emphasis, Idn. wavelet, Imc1.wavelet, large dependence high gray level emphasis.wavelet) (*p* < 0.001) were associated with LN regression. These results remained statistically significant in their validation cohort.

## 4. Discussion

This review explores the potential value of several markers to predict LARC response following neoadjuvant SCRT, TNT and RT. The 21 selected studies explored circulating biomarkers, tissue molecular biomarkers, tissue genomic biomarkers and MRI-derived radiomic markers. Statistically significant markers are summarized in Figure 2.

Markers of response to SCRT were studied in seven studies gathering four hundred and eighty three patients, showing significant results. In the pre-treatment setting, biomarkers associated with an improved tumor response were as follows: lower levels of CTLA4+ Treg, BCL-2 expression, high expression of Ku70 and MIB-1(Ki-67), low expression of nuclear β-catenin and MRI modifications (CBPFM Dp mean value and SD and VARPRO Dp mean value). In the time post-treatment but pre-surgery, reduction in the median value of cfDNA, higher levels of M-MDSC, lower levels of CTLA4+ Treg or PD1+ Treg and SIS changes on MRI were associated with an increased tumor response.

In TNT, six studies gathering four hundred and sixty eight patients found that pre-treatment markers associated with tumor response were as follows: low CEA levels, wild type TP53, SMAD4, KRAS and LRP1B, presence of the G-allele of LCS-6 and MRI features including short LN axis, smooth contour LN, homogeneous LN and Zhang et al. Rad score [35].

Finally, in alternative neoadjuvant RT regiments (eight studies gathering seven hundred and eighty five patients), significant pre-treatment predictive markers were as follows: low CEA circulating values, low hPEBP4 expression, high EGFR expression, low VEGF expression and wild type TP53.

Overall, the biomarker profile of responders to SCRT and RT supported an immune-permissive or immune-active tumor microenvironment. First, statistically significant markers included notably lower CTLA4+ Treg, which are known immunosuppressive cells that can repress the antitumor activity of the immune system [66]. Second, nuclear β-catenin plays a role in cell proliferation and immune evasion by inducing poor CD8+ T cells activity [51,52]. Third, hPEPB4 is associated with radioresistance [47,48]. Fourth, high expression of Ki67 and EGFR is associated with cell proliferation and high expression of BCL-2 is associated with cell survival [45,49]. These results correlate with the literature findings, where the immunosuppressive tumor microenvironment (TME) favorizes tumor growth, immune evasion and treatment resistance [67,68,69].

In TNT, however, responders had a less active tumor profile with lower CEA levels and without mutation in TP53, KRAS, LRP1B and SMAD4, which are usually associated with tumor proliferation [57,58,59]. This difference in the responder profile may reflect the potential effect of TNT on the TME. In fact, the impact of NAT may induce a modification of the TME, inducing immune activation, which can be correlated with improved tumor response and patient survival in RC and other solid tumor types [69,70,71,72].

These results should be interpreted with caution, as several limiting factors may impede their generalization. First, most studies had limited sample sizes (*n* = 12 to 240), lowering statistical power. Second, low reproducibility of the results may hinder their generalization. Most markers studied were assessed in single studies; only TP53 expression and mutations (nine articles), MMR status (four articles) and circulating CEA (three articles) have been replicated. Third, heterogeneity in the treatment options can be observed, especially in the TNT and in the alternative RT articles. Fourth, the lack of standardized assessment methods for tumors can lead to discrepancies in the interpretation of the results [13]. Some of the articles used the tumor regression grade with different scales, others used pCR and some used the TNM downstaging, sometimes considering only the T or the N downstaging. Fifth, the lack of standardized methods for the pre-analytic setting (sample type, sampling method, sample conservation, sampling timing, etc.) may influence studies results.

This review highlights the lack of translational research focusing on discovering predictive biomarkers to neoadjuvant SCRT, TNT or RT compared to the large number of studies focusing on predictive biomarkers for neoadjuvant CRT [14,15,65].

For instance, in the context of CRT, the mismatch repair (MMR) system and tumor-infiltrating lymphocytes (TILs) have emerged as biomarkers of interest for predicting tumor response and guiding treatment decisions [73,74,75]. Cercek and colleagues showed that MSI rectal tumors up to 25% are resistant to chemotherapy alone but tended to be sensitive to CRT [76]. Recently, several studies have shown that MSI LARC patients can probably be cured with anti-PD-1 immunotherapy alone [73,75].

The immune system plays a central role in tumor development, exhibiting both pro-tumor and anti-tumor activities [77]. In patients treated with neoadjuvant CRT, a high density of CD8+ TILs has been associated with an increased likelihood of achieving pathological complete response (pCR) [74,78]. Despite these promising findings, only four studies have investigated the predictive value of MMR status in non-CRT neoadjuvant settings [19,24,31,32], and only one study considered TILS in TNT without being able to reach statistical significance [32].

This limited evidence can be partially attributed to the relatively recent adoption of these alternative neoadjuvant regimens compared to CRT [79,80]. Additionally, the growing number of treatment protocols within NAT strategies poses challenges for conducting studies with sufficient sample sizes to yield meaningful conclusions.

## 5. Conclusions

LARC patient responders to neoadjuvant SCRT and RT tended to have a more immune-active tumor microenvironment, whereas responders to TNT tended to have a less active tumor profile. Although some biomarkers hold great promises, scarce publications, inconsistent results, low statistical power, and low reproducibility prevent them from reliably predicting tumor response following neoadjuvant SCRT, TNT or RT. This review highlights the urgent need for translational research with larger cohorts and standardized methods to tackle the challenge of discovering predictive markers of response to neoadjuvant treatments in patients with LARC treated with neoadjuvant SCRT, TNT or RT.

## Figures and Tables

**Figure 1 cancers-17-02229-f001:**
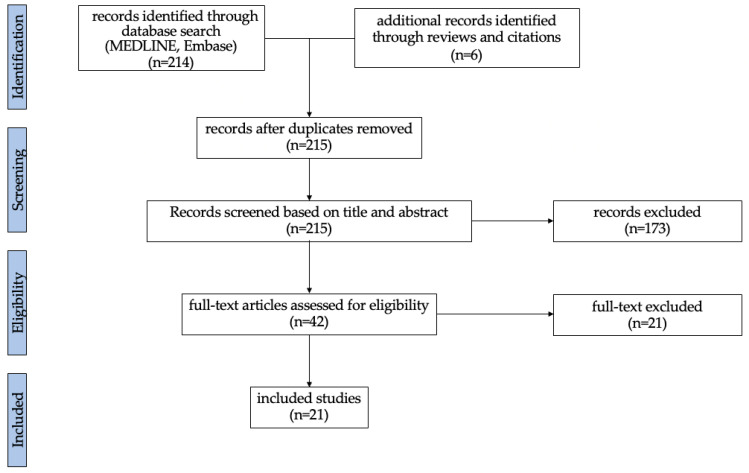
Flow chart illustrating the screening and selection process.

**Figure 2 cancers-17-02229-f002:**
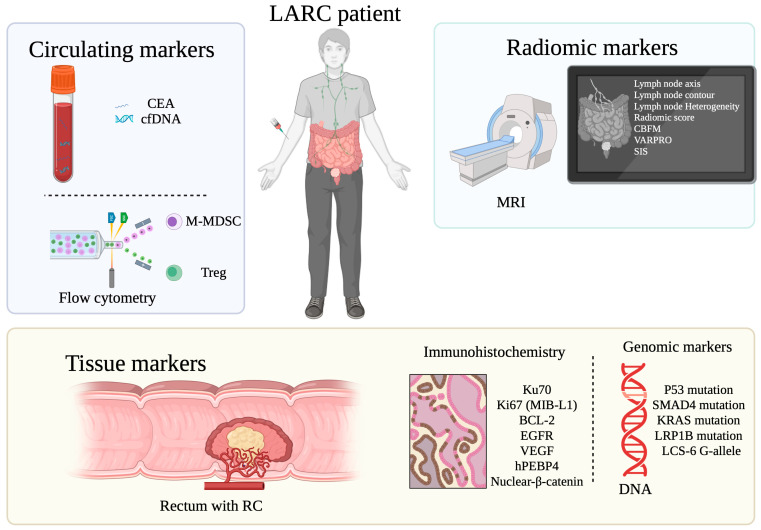
Overview of statistically significant biomarkers for tumor response prediction to SCRT, TNT and RT in LARC patients. Created in BioRender. Machado Carvalho, J. (2025) https://www.BioRender.com/seftlv5 (assessed on 18 April 2025).

**Table 1 cancers-17-02229-t001:** Overview of selected articles.

Authors	Study Year	Number of Patients	Treatment Type	Drugs and Radiation Dosage	Time to Surgery	Sampling Time	Technic	Measured Biomarker	Tumor Response Endpoint
Chapman et al. (2023) [31]	2016–2020	102	TNT (I + CRT)	FOLFOX or CAPOX + CRT (5-FU and 25–28 radiation fractions)	n/a	pre-treatment	Blood sample NGS	CEA MMR status 8 genes panel	cCR pCR
Fusco et al. (2019) [16]	2011–2016	34	SCRT	25 Gy in 5 fractions	4–6 weeks	pre-treatment post-treatment	Radiomic MRI	SIS IVIM DKI-derived parameters	TRG (Mandard)
Gasinska et al. (2011) [17]	2003–2006	122	SCRT	25 Gy in 5 fractions	1 or 4 weeks	pre-treatment post-treatment	FC	MIB-1 LI BrdUrd LI/SPF	CR (RECIST)
Gasinska et al. (2014) [18]	2003–2006	111	SCRT	25 Gy in 5 fractions	3–53 days	Pre-treatment	IHC	CD34/P53/BCL2/ Ku70/Ki-67/GLUT-1	CTR (RECIST) TRG (Dworak)
Ho et al. (2018) [19]	2000–2011	55	SCRT	25 Gy in 5 fractions	n/a	post-treatment	IHC	RAD50/MRE11/NBS1 complex MMR status	TRG (AJCC, 7th edition)
Iseas et al. (2021) [32]	2015–2018	36	TNT (I + CRT)	CAPOX + CRT (Capecitabin + 50.4 Gy in 28 fractions)	12–16 weeks	pre-treatment	IHC targeted DNA sequencing	MMR status TILs (CD3-CD8+)/HER2 PD-L1 72 genes panel	mrTRG TRG (CAP)
Kandioler et al. (2002) [23]	1994–1998	64	RT	25 Gy in 10 fractions	n/a	pre-treatment post-treatment	IHC targeted PCR	P53	T stage downstaging
Li et al. (2024) [36]	2020–2021	25	TNT (I + CRT + C)	CAPOX and camrelizmab + CRT +/− CAPOX	n/a	pre-treatment	NGS Liquid biopsy	ctDNA	pCR
Napolitano et al. (2015) [22]	n/a	13	SCRT	25 Gy in 5 fractions	8 weeks	pre-treatment post-treatment	FC IHC	G-MDSC M-MDSC Treg	TRG (Ryan)
Negri et al. (2008) [24]	n/a	38	RT	40 Gy in 16 fractions	6 weeks	pre-treatment	IHC	MMR status P53/p21/TS/VEGF	pCR
Qiu et al. (2013) [25]	2005–2010	86	RT	20 Gy in 5 fractions	4–5 weeks	pre-treatment post-treatment	IHC	hPEBP4	TRG (Dworak)
Rebischung et al. (2002) [26]	1989–1991	86	RT	39 Gy in 13 fractions	n/a	pre-treatment	Targeted DNA sequencing	p53	T downstaging
Saw et al. (2003) [27]	1991–1998	25	RT	50.4 Gy in 28 fractions	4–6 weeks	pre-treatment	IHC targeted PCR genotyping	p53/DCC/TS	TRG (Mandard)
Sclafani et al. (2015) [33]	2005–2008	155	TNT (Expert-C) (I + CRT)	Arm1 CAPOX + CRT (Capecitabin + 45 Gy in 25 fractions) Arm2 CAPOX +CRT +Cetuximab	4–6 weeks	pre-treatment	PCR genotyping	LCS6	pCR
Truelsen et al. (2022) [20]	2017–2020	12	SCRT	25 Gy in 5 fractions	6–8 weeks	pre-treatment during-treatment post-treatment	Blood sample Liquid Biopsy	cfDNA	TRG (Mandard)
Vidal et al. (2021) [34]	2015–2017	72	TNT (I + CRT)	Arm1 Aflibercept + mFOLOFX6 + CRT Arm2 mFOLFOX6 + CRT	n/a	pre-treatment post-treatment	Blood sample Liquid Biopsy	ctDNA	pCR
Wang et al. (2013) [21]	2008–2011	136	SCRT	25 Gy in 5 fractions	10–14 days	pre-treatment post-treatment	IHC	nuclear β-catenin	TRG (Dworak)
Wang et al. (2014) [28]	2003–2009	240	RT	30 Gy in 10 fractions	2–4 weeks	pre-treatment	Blood sample	CEA	T and N downstaging
Yao et al. (2014) [29]	2002–2005	142	RT	30 Gy in 10 fractions	14 days	pre-treatment	IHC Blood sample	HER2	TRG (AJCC, 7th edition)
Zhang et al. (2023) [35]	2019–2021	78	TNT (CRT + C)	CRT + mFOLFOX6 CRT + CapOx	8–10 weeks	post-treatment	Radiomic (MRI)	LN characteristics	LRG
Zlobec et al. (2008) [30]	n/a	104	RT	26 Gy in 4 fractions	6–8 weeks	pre-treatment	IHC	EGFR/VEGF/p53/ BCL-2/APAF-1	pCR

Legend**:** 5-FU: 5-Fluorouracil, ADC: apparent diffusion coefficient, AJCC: American joint committee on cancer, APAF-1: apoptotic protease activating factor-1,BrdUrdLI: bromodeoxyuridine labeling index, C: consolidation therapy, CAPOX: capecitabin and oxaliplatin, CD: cluster of differentiation, CEA: carcinoembryonic antigen, cfDNA: cell-free DNA, CRT: chemoradiotherapy, ctDNA: circulating tumor DNA, CR: complete response, cCR: clinical complete response, DCC: deleted in colorectal cancer, DKI: diffusion kurtosis imaging, DNA: deoxyribonucleir acid, EGFR: epidermal growth factor receptor, FC: flow cytometry, FOLFOX: folinic acid, 5-fluorouracil and oxaliplatin, GLUT-1: glucose transporter 1, G-MSDC: granulocytic myeloid-derived suppressor cells, HER2: receptor tyrosine–protein kinase erb-2, hPEBP4: human epidermal growth factor receptor-2, IHC: immunohistochemistry, I: induction therapy, IVIM: intravoxel incoherent motion, LN: lymph node, LRG: lymph node regression grade, mFOLFOX6: modified folinic acid, 5-fluorouracil and oxaliplatin, MIB-LI: MIB labeling index, M-MDSC: monocytic myeloid-derived suppressor cells, MMR: mismatch repair, MP: mucin pool, MRI: magnetic resonance imaging, mrTRG: tumor regression grade using MRI, N: lymph node stage according to the TNM classification, NGS: next-generation sequencing, n/a = not available, PCR: polymerase chain reaction, pCR: pathological complete response, RECIST: Response Evaluation Criteria in Solid Tumors, RT: radiotherapy, SCRT: short-course radiotherapy, SIS: standardized index of shape, SPF: S-phase fraction, T: tumor size according to the TNM classification, TILs: tumor infiltrating lymphocytes, TNT: total neoadjuvant treatment, Treg: regulatory T cells, TRG: tumor regression grade, TS: thymidylate synthase, VEGF: vascular growth factor.

## Data Availability

No new data were created or analyzed in this study. Data sharing is not applicable to this article.

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
