# Peer review of "Narrative Review: Predictive Biomarkers of Tumor Response to Neoadjuvant Radiotherapy or Total Neoadjuvant Therapy of Locally Advanced Rectal Cancer Patients"

_cancers, 2025, doi:10.3390/cancers17132229_

Round 1
Reviewer 1 Report
Comments and Suggestions for Authors
The review presents a literature search of studies focusing on predictive biomarkers of response in LARC patients treated with radiotherapy alone (RT), short-course radiotherapy (SCRT), or total neoadjuvant therapy (TNT). The latter combines upfront chemotherapy followed or preceded by either long-course chemoradiotherapy (CRT) or SCRT, and represents a recent paradigm shift in the management of LARC. Studies using the standard long-course CRT were deliberately excluded, as this approach has already been extensively investigated. However, this choice is highly questionable, as the review and comparison of studies is incomplete: key biomarkers involved in the response to CRT may also be relevant for TNT. Furthermore, the review includes very outdated studies that consider obsolete approaches such as radiotherapy alone, as well as SCRT, which is currently used in only a few countries.
Author Response
Comment 1: The review presents a literature search of studies focusing on predictive biomarkers of response in LARC patients treated with radiotherapy alone (RT), short-course radiotherapy (SCRT), or total neoadjuvant therapy (TNT). The latter combines upfront chemotherapy followed or preceded by either long-course chemoradiotherapy (CRT) or SCRT, and represents a recent paradigm shift in the management of LARC. Studies using the standard long-course CRT were deliberately excluded, as this approach has already been extensively investigated. However, this choice is highly questionable, as the review and comparison of studies is incomplete: key biomarkers involved in the response to CRT may also be relevant for TNT. Furthermore, the review includes very outdated studies that consider obsolete approaches such as radiotherapy alone, as well as SCRT, which is currently used in only a few countries.
Response 1: Thank you very much for your comment and for the time you dedicated in reviewing our article.
We would like to offer several clarifications regarding the scope and rational of our review.
First, we fully acknowledge that some key biomarkers involved in the response to CRT, such as tumor infiltrating lymphocytes or the mismatch repair system, may also be relevant for patients undergoing TNT and SCRT. We were, in fact, surprised to find that only a limited number of studies have investigated these biomarkers in the context of neoadjuvant non-CRT. By highlighting this gap, our aim was to underscore the scarce literature available and the need of further research. A dedicated paragraph was added to the discussion (p.11, ll. 365-377).
Second, while it is true that neoadjuvant radiotherapy alone is being less frequently chosen in the treatment of LARC patients, SCRT remains a recognized alternative to CRT and still continues to be recommended in NCCN and ESMO guidelines, including the 2025 updates.
Third, regarding articles considering other radiotherapy regiments considered as obsolete nowadays, we decided to include them for two reasons. First, even if these regiments are not currently used in clinical practice, biomarkers explored in these studies, as VEGF, HER2 or EGFR could remain of clinical interest. Second of all, the inclusion of these articles emphasizes the scarce literature regarding biomarkers predictive of tumor response in LARC patients undergoing non-CRT neoadjuvant treatments, with only 21 articles retrieved despite large inclusion criteria.
Reviewer 2 Report
Comments and Suggestions for Authors
The authors of the manuscript have collated some of the available published data on predictive markers of tumor response to neoadjuvant treatment of locally advanced rectal cancers. They have correlated the published data on both tissue and imaging-based biomarkers and have identified factors that can potentially predict for better response to treatment.
A few additional points that can make the manuscript more comprehensive will be:
- PET/CT as a tool for evaluation of response has not been described in the manuscript. Was this intentional? I believe reporting on this will be appropriate since the authors have talked about radiomics.
- Can the authors comment on MMR/MSI specifically? There is significant data that MMR deficient tumor response is inferior to chemoRT when compared to MMR proficient tumors. This all the more relevant with the recent data that shows a significant response of MMR def tumors to immunotherapy.
- While pathological response was the primary outcome assessed in this study, can the authors comment briefly about any outstanding biomarker profile that they found that correlates to survival outcomes?
Author Response
Comment 1: PET/CT as a tool for evaluation of response has not been described in the manuscript. Was this intentional? I believe reporting on this will be appropriate since the authors have talked about radiomics.
Response 1: First of all, thank you very much for reviewing our article. Considering your first comment that got all our attention, we would like to point out that we did not exclude articles considering PET-CT, CT-scans or other radiological imaging. However, the two retrieved articles considering the predictive value of radiomic features in LARC patients undergoing non-CRT neoadjuvant treatment performed MRI. We did not find articles using CT or PET-CT in this setting.
Comment 2: Can the authors comment on MMR/MSI specifically? There is significant data that MMR deficient tumor response is inferior to chemoRT when compared to MMR proficient tumors. This all the more relevant with the recent data that shows a significant response of MMR def tumors to immunotherapy.
Response 2: Thank you very much for this comment. The MMR system is indeed of high interest in RC with recent results encouraging the use of immunotherapy for patients with microsatellite instability. We were surprised to find that only four studies considered its predictive value in LARC patients undergoing neoadjuvant RT, SCRT or TNT. Despite the non-significant results in these articles, we decided to further develop this topic (p.11, ll 365-377).
Comment 3: While pathological response was the primary outcome assessed in this study, can the authors comment briefly about any outstanding biomarker profile that they found that correlates to survival outcomes?
Response 3: Thank you for this insightful question.
Unfortunately, survival outcomes such as overall survival or disease-free survival were beyond the scope of our review, which focused specifically on biomarkers predictive of pathological response. However, we recognize the clinical importance of identifying biomarkers with prognostic value.
Reviewer 3 Report
Comments and Suggestions for Authors
The concept of the review is interesting but the data included are limited. The manuscript needs thorough revision and expansion to include additional important markers.
Suggestions are as follows:
- Page 7: Please report the exact lymphocyte markers applied in the flow cytometry study.
- Plasma cytokines are ignored.
- The Table should contain 21 studies.
- Page 7, 3.1 Immunohistochemistry: Please report the subcellular patterns (e.g. membrane or cytoplasmic staining) that authors used to assess specimens and also the cut-off points used to identify groups of high vs. low expression
- The role of Tumor infiltrating lymphocytes and specific lymphocyte subset density has been ignored in the paper. The same applies for immune checkpoint molecules, MMR genes and so on.
- Angiogenesis, hypoxia and metabolism markers are ignored.
Author Response
Comment 1: Page 7: Please report the exact lymphocyte markers applied in the flow cytometry study.
Response 1: Thank you very much for your comment. The leukocyte markers have been added next to the first mention of the cell lines in page 7.
Comment 2: Plasma cytokines are ignored.
Response 2: Thank you for pointing it out. We would like to confirm that plasma cytokines or any other biomarkers have not been intentionally excluded from our review. Plasma cytokines could not be discussed as none of the retrieved articles considered their predictive value in LARC patients undergoing non-CRT neoadjuvant treatment.
Comment 3: The Table should contain 21 studies.
Response 3: We double-checked the table and confirm that it contains 21 articles.
Comment 4: Page 7, 3.1 Immunohistochemistry: Please report the subcellular patterns (e.g. membrane or cytoplasmic staining) that authors used to assess specimens and also the cut-off points used to identify groups of high vs. low expression
Response 4: Thank you for pointing it out. We revised our article, accordingly, specifying the subcellular patterns and cut-off for statistically significant markers studied in immunohistochemistry (pp. 7-8).
Comment 5: The role of Tumor infiltrating lymphocytes and specific lymphocyte subset density has been ignored in the paper. The same applies for immune checkpoint molecules, MMR genes and so on.
Response 5: Thank you for your valuable comment.
As mentioned in our response to Comment 2, no biomarkers were intentionally excluded from our review. Regarding the specific biomarkers you mentioned—tumor-infiltrating lymphocytes, immune checkpoint molecules, and MMR genes—we did identify a limited number of relevant studies within the context of non-CRT neoadjuvant treatments in LARC patients: one article addressed tumor-infiltrating lymphocytes, four articles examined MMR status, and one investigated PD-L1 expression (the only immune checkpoint molecule reported).
However, these studies did not demonstrate statistically significant predictive value, which is why they were not discussed in detail in the results section (as noted on page 8). That said, we fully agree that these biomarkers remain of considerable interest given their established relevance in other treatment contexts. In recognition of their potential importance, we have added a dedicated section in the Discussion (page 11, ll. 365-377) to elaborate on their relevance and highlight the need for further research.
Comment 6: Angiogenesis, hypoxia and metabolism markers are ignored.
Response 6: Thank you again for this comment. As for the second and fifth comment, we would like to ensure again that no biomarker was intentionally omitted. We would also like to point out that regarding angiogenesis and hypoxia specifically, VEGF was one of the statistically significant markers retrieved and discussed in our review (page 8).
Round 2
Reviewer 1 Report
Comments and Suggestions for Authors
The authors replied to comments and attempted to justify their decision to exclude studies evaluating predictive biomarkers for standard chemoradiotherapy from the review, highlighting a gap in the literature specifically concerning translational research on treatment regimens such as SCRT and TNT.